# A bioengineered niche promotes in vivo engraftment and maturation of pluripotent stem cell derived human lung organoids

**Briana R Dye[1], Priya H Dedhia[2], Alyssa J Miller[3,4], Melinda S Nagy[3], Eric S White[3], Lonnie D Shea[5,6], Jason R Spence[1,3,5]***

[1]Department of Cell and Developmental Biology, University of Michigan Medical School, Ann Arbor, United States; [2]Department of Surgery, University of Michigan Medical School, Ann Arbor, United States; [3]Department of Internal Medicine, University of Michigan Medical School, Ann Arbor, United States; [4]Program in Cellular and Molecular Biology, University of Michigan Medical School, Ann Arbor, United States; [5]Center for Organogenesis, University of Michigan Medical School, Ann Arbor, United States; [6]Biomedical Engineering, University of Michigan Biomedical Engineering, Ann Arbor, United States

**Abstract** Human pluripotent stem cell (hPSC) derived tissues often remain developmentally immature in vitro, and become more adult-like in their structure, cellular diversity and function following transplantation into immunocompromised mice. Previously we have demonstrated that hPSC-derived human lung organoids (HLOs) resembled human fetal lung tissue in vitro (*Dye et al., 2015*). Here we show that HLOs required a bioartificial microporous poly(lactide-co-glycolide) (PLG) scaffold niche for successful engraftment, long-term survival, and maturation of lung epithelium in vivo. Analysis of scaffold-grown transplanted tissue showed airway-like tissue with enhanced epithelial structure and organization compared to HLOs grown in vitro. By further comparing in vitro and in vivo grown HLOs with fetal and adult human lung tissue, we found that in vivo transplanted HLOs had improved cellular differentiation of secretory lineages that is reflective of differences between fetal and adult tissue, resulting in airway-like structures that were remarkably similar to the native adult human lung.

**\*For correspondence:** spencejr@umich.edu

**Competing interests:** The authors declare that no competing interests exist.

## Introduction

The past decade has seen an exciting emergence of new models of human development and disease (*St Johnston, 2015*). These advances stem from our ability to culture primary human tissues and derive complex three-dimensional organ-like tissues, called organoids from human pluripotent stem cells (hPSCs) (*Dedhia et al., 2016*; *Fatehullah et al., 2016*; *Johnson and Hockemeyer, 2015*; *Huch and Koo, 2015*; *Rookmaaker et al., 2015*; *Dye et al., 2016*). Both tissue-derived and hPSC-derived human lung models have been developed and recapitulate some structural and cellular features of the human lung (*Barkauskas et al., 2013*; *Rock et al., 2009*; *Gotoh et al., 2014*; *Konishi et al., 2016*; *Vaughan et al., 2006*; *Pageau et al., 2011*; *Fessart et al., 2013*; *Franzdóttir et al., 2010*; *Kaisani et al., 2014*; *Dye et al., 2015*; *Firth et al., 2014*; *Mou et al., 2016*). For example, we recently described methods to direct differentiation of hPSCs in a stepwise process, which mimicked aspects of in vivo lung development, into three-dimensional human lung organoids (HLOs). HLOs possessed closed epithelial cysts resembling airway-like structures, which

consisted of epithelial cells primarily expressing ciliated and basal cell markers. The airway-like structures were surrounded by mesenchyme, including cells expressing smooth muscle and myofibroblast markers (*Dye et al., 2015*). Despite having airway-like organization around a lumen, the HLO epithelium was disorganized and did not possess some cell types found in the mature airway, including club and goblet secretory cell lineages. Consistent with in vitro differentiated HLOs being immature, transcriptome-wide comparisons showed that HLOs were similar to the human fetal lung (*Dye et al., 2015*).

In addition to HLOs, intestinal, kidney and cerebral organoids are similar to the analogous human fetal organ (*Dye et al., 2015*; *Finkbeiner et al., 2015b*; *Camp et al., 2015*; *Takasato et al., 2015*). Human intestinal organoids (HIOs), for example, are an in vitro three-dimensional model of the human intestine derived from hPSCs that closely resemble the fetal intestine. When transplanted under the mouse kidney capsule, HIOs mature and gain relevant adult structures including villi and crypts (*Finkbeiner et al., 2015b*; *Watson et al., 2014*; *Finkbeiner et al., 2015a*). In the current work, our goal was to transplant HLOs to determine if an in vivo environment would enhance the structural organization and cellular differentiation of the airway-like tissue.

Our results show that, unlike intestinal organoids, HLOs placed under the mouse kidney capsule, in the omentum, or the epididymal fat pad did not generate lung epithelial structures or cell types. Thus, while all of these highly vascular sites have been utilized for engraftment in other contexts (*Finkbeiner et al., 2015b*; *Watson et al., 2014*; *Pepper et al., 2013*; *Bartholomeus et al., 2013*; *Smink et al., 2013*), HLOs did not thrive in these environments. We turned to microporous poly(lactide-co-glycolide) (PLG) scaffolds in order to create an alternative niche for the HLOs during transplantation since PLG scaffolds have led to improved survival and function of pancreatic beta cells following transplantation (*Gibly et al., 2013*; *Blomeier et al., 2006*; *Hlavaty et al., 2014*; *Graham et al., 2013*; *Kheradmand et al., 2011*).

Our results demonstrate that microporous PLG scaffolds provided a niche for HLOs that enhance survival and engraftment upon transplantation into the epididymal fat pad of NOD-*scid* IL2Rgnull (NSG) mice. After 8–15 weeks, the retrieved transplanted HLOs (tHLOs) possessed airway-like structures with improved epithelial organization resembling the human adult lung and demonstrated enhanced cellular differentiation into basal, ciliated, club, and goblet cells. The tHLO airway structures were vascularized, and surrounded by mesenchymal cells that expressed both smooth muscle and myofibroblast markers, in addition to areas of organized cartilage. This work demonstrates that hPSC-derived lung tissue can give rise to complex multicellular airway-like structures in vivo, similar to those found in the adult human lung.

## Results

### Lung epithelium does not persist when HLOs are transplanted into mice

It has been shown that hPSC derived intestinal organoids acquire crypt and villus structures resembling the adult intestine along with mature cell types by transplantation into a highly vascular in vivo environment such as the kidney capsule or the abdominal omentum (*Finkbeiner et al., 2015b*; *Watson et al., 2014*). A similar strategy was employed in an attempt to engraft and mature HLOs, in which several different experimental conditions and engraftment sites were attempted utilizing NSG mice. Experiments were initially conducted using the hESC line UM63-1, and all major findings were reproduced in two additional hESC lines; H1 and H9 (*Table 1*). Data presented throughout the manuscript are from the hESC line UM63-1, unless otherwise stated. In our first attempt, 35d (35 day old) HLOs were placed under the kidney capsule and were harvested after 4 weeks (*Figure 1—figure supplement 1A–B*). The retrieved organoids expressed the human-specific mitochondria marker (huMITO), but lacked NKX2.1+ lung epithelium (*Table 1*, *Figure 1—figure supplement 1A–C*). We hypothesized that an earlier stage of HLO cultures may be more proliferative and therefore have better survival upon engraftment. 1d HLOs were injected under the kidney capsule (*Table 1*, *Figure 1—figure supplement 1D*). After 6 weeks, the tissue had expanded, surpassing the size of the kidney (*Figure 1—figure supplement 1E*). Further analysis demonstrated that the tissue was of human origin (huMITO+), but no NKX2.1+ epithelium was observed (*Figure 1—figure supplement 1F*). Thus, the age of transplanted HLOs did not seem to affect the survival of the HLO lung epithelium.

**Table 1.** Overview of Organoid transplants. Transplant site refers to where the tissue was placed in the mouse. HLOs grown in vitro from 1 to 65 days (d) were transplanted and tissues were harvested at various time points ranging from 4 to 15 weeks (wks). Three hESC lines were used including UM63-1, H9, and H1. The most successful transplants that contained mature airway-like structures were 1d HLOs seeded onto the PLG scaffolds with or without Matrigel and FGF10 after 8 to 15 weeks.

| Transplant site | Transplanted tissue | Time | N | Treatment | Procedure | Outcome | Cell line |
|---|---|---|---|---|---|---|---|
| Kidney Capsule | 35d HLOs | 4 weeks | 6 | - | Placed with forceps | 6/6 huMITO+ NKX2.1- | UM63-1 |
| Kidney Capsule | 1d HLOs | 6 weeks | 3 | Mixed with 100% Matrigel | Injected into capsule | 3/3 huMITO+ NKX2.1- | UM63-1 |
| Omentum | 65d HLOs | 12 weeks | 13 | - | Sutured into greater omentum | 11/13 huMITO+ NKX2.1- 2/13 huMITO+ NKX2.1- Immature airway-like structures | UM63-1 |
| Fat Pad | 1d HLOs | 8 weeks | 5 | 100% Matrigel plug filled with spheroids | Enveloped by epididymal fat pad | No tissue retrieved | H9 |
| Fat Pad | 1d HLOs | 4 weeks | 4 | Seeded on Scaffold mixed with 100% Matrigel and FGF10 | Enveloped by epididymal fat pad | 4/4 huMITO+ NKX2.1+ immature airway-like structures | UM63-1 |
| Fat Pad | 1d HLOs | 8 weeks | 8 | Seeded on scaffold, mixed with 100% Matrigel and FGF10 | Enveloped by epididymal fat pad | 8/8 huMITO+ NKX2.1+ mature airway-like structures | UM63-1 |
| Fat Pad | 1d HLOs | 8 weeks | 4 | Seeded on scaffold (without Matrigel, FGF10) | Enveloped by epididymal fat pad | 4/4 huMITO+ NKX2.1+ mature airway-like structures | UM63-1 |
| Fat Pad | 1d HLOs | 15 weeks | 3 | Seeded on scaffold (without Matrigel, FGF10) | Enveloped by epididymal fat pad | 3/3 huMITO+ NKX2.1+ mature airway-like structures | H9 |
| Fat Pad | 1d HLOs | 8 weeks | 4 | Seeded on Scaffold (without Matrigel, FGF10) | Enveloped by epididymal fat pad | 4/4 huMITO+ NKX2.1_ mature airway-like structures | H1 |

Next, we assessed the effect of the transplant site on HLO maturation. 65d HLOs were placed into the abdominal omentum and secured in place with a stitch, which also allowed us to identify the site of the transplant. Tissues were harvested after 12 weeks (*Table 1*, *Figure 1—figure supplement 1G–H*), and again were positive for huMITO, but the majority did not have evidence of a lung epithelium (*Figure 1—figure supplement 1I*). Only 2 of the 13 mice transplanted in this cohort demonstrated huMITO+ airway-like structures as indicated by expression of the lung epithelial marker NKX2.1, the basal cell marker P63, and the ciliated cell marker FOXJ1 (*Figure 1—figure supplement 2*). Taken together, these data indicated that alternative sites of transplantation did not robustly support survival or growth of the HLO lung epithelium in vivo (*Table 1*).

## Microporous scaffolds provide a niche enhancing in vivo engraftment and survival of lung epithelium

Given that we recovered human tissues that were largely devoid of lung epithelium, we hypothesized that transplantation of HLOs would be enhanced if provided a structural niche, which has been demonstrated to improve engraftment, vascularization and to support the survival of transplanted pancreatic beta cells (*Gibly et al., 2013*; *Blomeier et al., 2006*; *Hlavaty et al., 2014*; *Graham et al., 2013*; *Kheradmand et al., 2011*; *Gibly et al., 2011*). Microporous PLG scaffolds provide a rigid environment for the tissue to adhere to along with a porous (250–425 μm diameter) honeycomb-like structure to enable tissue growth and infiltration of vasculature (*Figure 1A*). In order to prepare PLG-HLO constructs for transplant, HLOs were suspended in Matrigel and pipetted onto the scaffolds. 1d HLOs were able to adhere to the pores on the scaffold, with the majority of the HLOs scattered across the 5 mm diameter surface, and a few towards the middle of the 2 mm thick scaffold (*Figure 1B–C*). HLO-seeded PLG scaffolds were subsequently cultured for 5 to 7 days in vitro submersed in 500 ng/mL FGF10 supplemented media, the same media used to grow HLOs in vitro (*Dye et al., 2015*) (*Figure 1D*). Following 5–7 days of culture, constructs were transplanted directly, or dipped in Matrigel supplemented with FGF10 (500 ng/mL) prior to transplantation. Both conditions were transplanted into the mouse epididymal fat pad (Matrigel/FGF10 n = 8; without

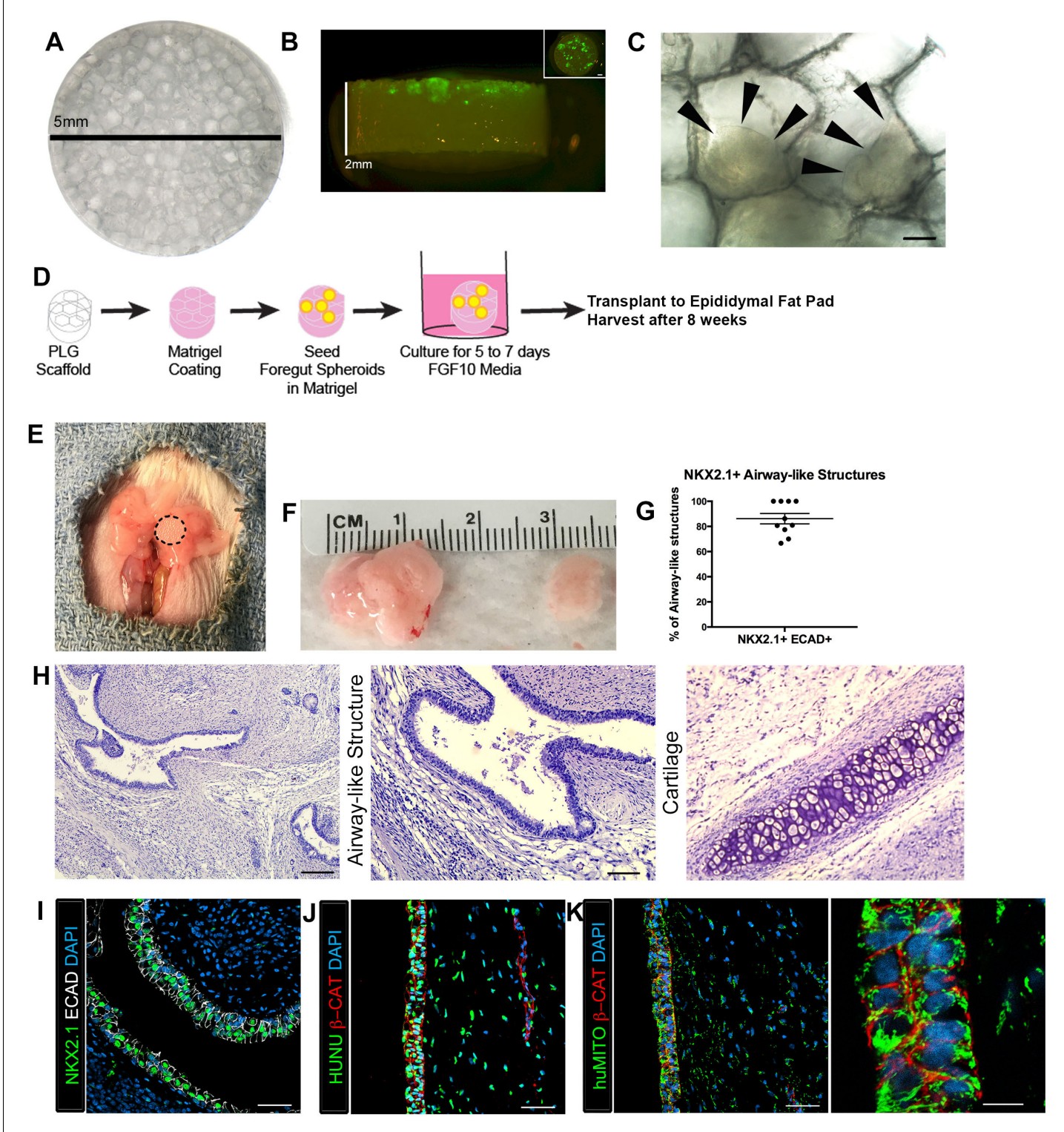

**Figure 1.** Transplanted HLO-scaffold constructs engrafted, grew and possessed airway-like structures. (A) PLG scaffold are 5 mm in diameter with honeycomb-patterned architecture. (B) The majority of Di-O labeled 1d HLOs (green) remained at the surface of the scaffold with a few organoids descending toward the middle of the scaffold. Inset shows aerial view of the scaffold with 1d HLOs (green) scattered throughout. (C) 1d HLOs settled within the pores of the scaffold. Scale bar represents 100 μm. (D) PLG scaffolds were seeded with 1d HLOs and cultured for 5 to 7 days in vitro in media supplemented with FGF10. The HLO-laden scaffolds were then transplanted into the mouse epididymal fat pad and harvested at 8 weeks. (E) HLO-scaffold (dotted line) was placed in mouse epididymal fat pad. (F) Transplanted HLOs (tHLOs) ranged from 0.5 cm to 1.5 cm in length. (G) The average

*Figure 1 continued on next page*

*Figure 1 continued*

number airway-like structures that were NKX2.1+ ECAD+ out of all ECAD+ structures was 86.19% +/- 4.14% (N = 10, error bars represent SEM). (**H**) H&E of tHLOs showed airway-like structures (right two panels, low and high mag) and pockets of cartilage (left panel). Scale bar at low mag represents 200 µm and high mag 100 µm. (**I**) Airway-like structures outlined by ECAD (white) expressed the lung marker NKX2.1 (green). Scale bar represents 50 µm. (**J–K**) Both the epithelium (β-CAT, red) and mesenchyme expressed the human nuclear marker, HUNU (**J**, green) and the human mitochondrial marker huMITO (**K**, green). Scale bars represent 50 µm in **J–K** and 10 µm in high mag image in **K**.

The following source data and figure supplements are available for figure 1:

**Source data 1.** Summary of NKX2.1+ epithelial structures in individual tHLOs.

**Figure supplement 1.** Highly vascular engraftment sites did not maintain lung epithelium.

**Figure supplement 2.** HLO omentum transplants maintained lung epithelium poorly.

**Figure supplement 3.** 1d HLOs grown on a scaffold and transplanted into the mouse epididymal fat pad expressed lung markers when harvested at 4 weeks.

**Figure supplement 4.** Transplanted HLO-scaffold constructs retrieved 4 weeks post-transplantation possessed airway-like structures that expressed basal and ciliated cell markers.

**Figure supplement 5.** 1d HLOs grown on PLG scaffolds in vitro maintained NKX2.1 expression but did not generate airway-like structures after 8 weeks.

Matrigel/FGF10 n = 4, *Table 1* and *Figure 1D–E*). In both conditions, 100% of the recovered constructs possessed NKX2.1+ airway-like structures (n = 12 total) (Characterized in *Figures 1–3*, *Table 1*), suggesting that the addition of Matrigel/FGF10 prior to transplant was not necessary, but that the scaffold provided critical support for engraftment and survival of the lung epithelium. The epididymal fat pad was chosen as the site for transplantation because it was the only site able to accommodate the large size of the scaffold-HLO construct while still providing a highly vascular environment. As a control, 1d HLOs in a Matrigel plug (without being placed on a scaffold) were transplanted into the epididymal fat pad but no tissue was recovered at 8 weeks post-transplant (*Table 1*). We also demonstrated that HLOs generated from three independent hESC lines (H1, H9, UM63-1) developed similarly following transplantation on scaffolds (*Table 1*). Collectively, these data demonstrated that PLG scaffolds provide a robust niche, enabling HLOs to engraft and survive in the epididymal fat pad.

The Matrigel/FGF10 treated scaffold constructs were harvested at two time points, 4 weeks and 8 weeks after transplant (*Table 1*). After 4 weeks of transplantation, constructs had airway-like structures that resembled in vitro grown HLOs (*Figure 1—figure supplement 3*) (*Dye et al., 2015*). 4 week HLO constructs had NKX2.1 +/huMITO+ airway-like structures that consisted of cells expressing the basal cell marker P63 and ciliated markers FOXJ1 and Acetylated Tubulin (ACTTUB), but the secretory club cell marker CC10 (also known as SCGB1A1) was not detected, suggesting that 4 weeks of transplantation did not lead to enhanced structure within HLOs (*Figure 1—figure supplement 4*).

The scaffold-HLO constructs harvested after 8 weeks ranged in size from 0.5 cm to 1.5 cm (*Figure 1F*). We observed that each transplanted HLO had multiple epithelial structures per cross section. To further characterize the in vivo grown HLOs, we quantified the total number of epithelial structures (ECAD+) per cross-section and found that 86.19% of all ECAD structures were also NKX2.1+ (*Figure 1G,I*, *Figure 1—source data 1*). The histology of the transplanted tissue revealed airway-like structures tightly surrounded by mesenchymal cells. In addition, there were pockets of organized cartilage throughout the transplants (*Figure 1H*). Epithelial structures were verified to be of human origin using huMITO and the human-specific nuclear marker HUNU (*Figure 1J–K*). As a control, 1d HLOs were seeded on the scaffold and grown in vitro for 4 to 8 weeks. These in vitro grown HLO-scaffolds had significantly less growth, and although some cells expressed the lung marker NKX2.1, no airway structures were observed (*Figure 1—figure supplement 5*). This is

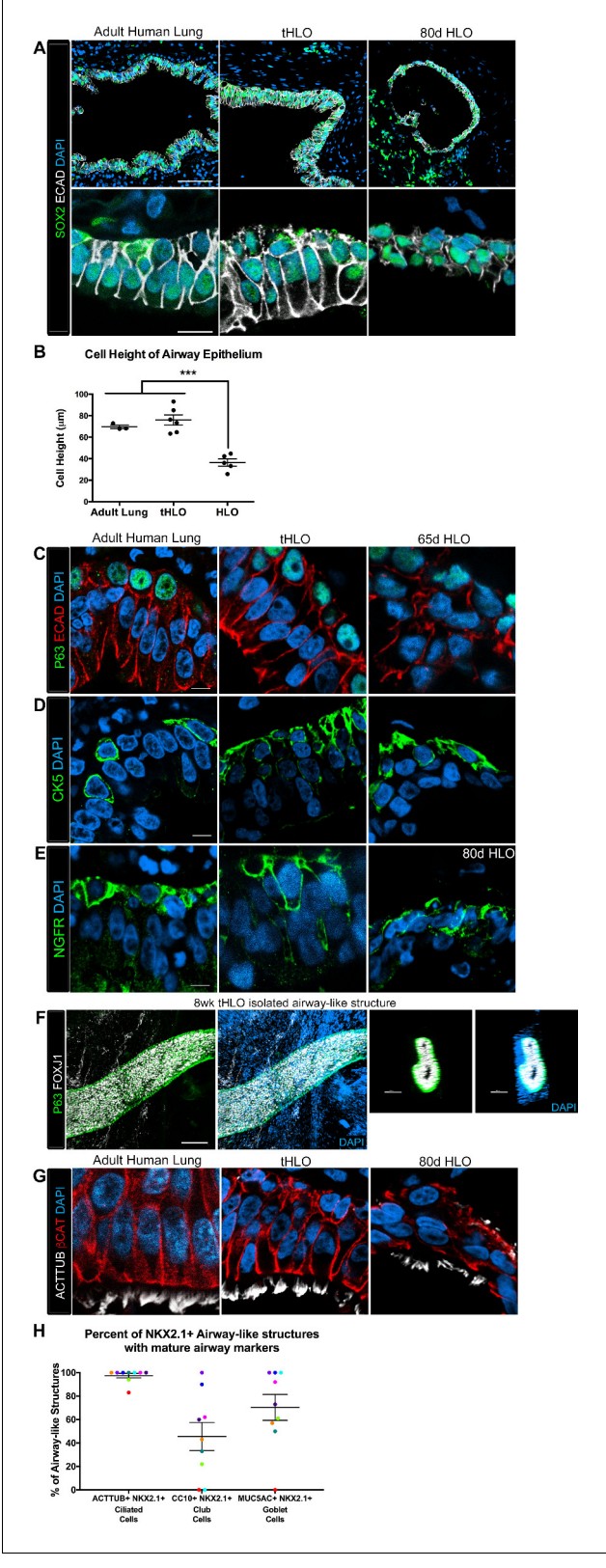

**Figure 2.** Transplanted HLO-scaffold constructs harvested at 8 weeks possessed mature airway-like structures and had an enhanced epithelial structure. (**A**) Adult human lung, tHLOs and 80d HLOs possess SOX2+ (green) epithelium marked by ECAD (white). Only adult lung airways and tHLO airway-like structures possessed a pseudostratified epithelium (ECAD, white). Scale bars represent 50 μm in low mag images and 10 μm in high mag. *Figure 2 continued on next page*

*Figure 2 continued*

(B) Measurements of cell height were taken from adult human lung (n = 3), tHLO (n = 5), and HLO (n = 6) airways of cells facing toward the lumen. Averages were adult: 69.59 μm ± 1.65, tHLO: 75.69 μm ± 4.74, HLO = 36.39 μm ± 3.39. *** represents p<.0005 and error bars represent SEM. All HLOs were derived from hESC line UM63-1. (C–E) Adult human lung, tHLOs, and HLOs (65d, 80d) expressed the basal cell markers P63 (C, green), cytokeratin5 (CK5, D, green), and NGFR (E, green) Scale bars represent 10 μm. (F) 3D rendering of z-stack images revealed tube-like structures with cells lining the tube expressing the basal cell marker P63 (green) and cells within the tube cells expressing the ciliated cell marker FOXJ1 (white). A cross section of the z-stack images through the tube revealed that P63 (green) lines the tube while FOXJ1+ cells (white) are within the tube. Scale bars represent 100 μm. (G) Adult human lung, tHLO, and 80d HLO possessed ciliated cells labeled by ACTTUB (white) with the cilia facing in toward the lumen. Scale bars represent 10 μm. (H) NKX2.1+ airway-like structures within each tHLO (n = 9) that contained NKX2.1+ACTTUB+(ciliated cells), NKX2.1+CC10+ (club cells), or NKX2.1+MUC5AC+ (goblet cells) in each tHLO. Data was quantified from UM63-1 hESC-derived tHLOs transplanted for 8 weeks. Each independent tHLO counted is represented by a different color. (n = 9, error bars represent SEM).

The following source data and figure supplement are available for figure 2:

**Source data 1.** Summary of NKX2.1+ airway-like structures that contain ACTTUB+ cells, CC10+ cells, or MUC5AC + cells in individual UM63-1 hESC-derived tHLOs.

**Figure supplement 1.** Human airways, in vitro grown HLOs, and transplanted HLOs express airway markers.

---

possibly due to the fact that HLOs grown in vitro are normally embedded in Matrigel and passaged every two weeks whereas tissue grown on the scaffolds could not be passaged, and were not embedded in Matrigel. Thus, results from in vitro grown scaffold-HLO constructs suggest that the scaffold alone does not support long term HLO growth or induce the enhanced epithelial organization or maturation, but the combination of the scaffold and the in vivo environment allows for growth and maturation of the HLO epithelium.

## In vivo grown HLO-scaffold constructs have enhanced epithelial structure and cellular differentiation

The native lung is organized into airway structures and alveoli. Interestingly, even though in vitro grown HLOs possessed alveolar cell types (*Dye et al., 2015*), we could not find evidence for the ATI marker HOPX or the ATII marker SFTPC in tHLOs at 4 or 8 weeks indicating that these cell types do not persist in vivo (data not shown). Native lung airways are organized into a SOX2+ pseudostratified epithelium with basal and multiciliated cells being the most prevalent cell types in the airway (*Mercer et al., 1994*; *Li et al., 2013*; *Morrisey and Hogan, 2010*). Both non-transplanted HLOs and transplanted HLOs expressed the airway marker SOX2; however, tHLOs on scaffolds formed a highly organized pseudostratified epithelium, including taller ECAD+ cells that had a clearly visible apical and basal surface similar to the adult human airway (*Figure 2A*; Note that adult human airways examined are consider lower bronchi as detailed in the Materials and methods). The average cell height – from the apical to basal surface – of cells lining the tHLO airway-like lumen was similar to adult lungs while the non-transplanted HLOs grown in Matrigel had significantly shorter cells than the tHLOs and adult lungs (*Figure 2B*). In both fetal upper airways and adult airways, basal cells line the basal side of the airway epithelium and express markers including P63, cytokeratin 5 (CK5), and NGFR (2C-E, *Figure 2—figure supplement 1A–C*; Note that fetal human airways examined are considered upper bronchi, as detailed in the Materials and methods). Both non-transplanted and transplanted HLOs possessed cells lining the basal side of the airway-like structures that expressed the basal cell markers P63, CK5, and NGFR, however; basal cells in tHLOs had cellular organization and shape more reminiscent of the native human lung (*Figure 2C–E*).

Upon dissection of tHLOs, we observed tube-like structures within the 8 week constructs. Microdissected tube-like structures were stained for FOXJ1 and P63 using whole-mount immunostaining, followed by confocal imaging (*Figure 2F*). Whole-mount imaging revealed tube-like structures with abundant FOXJ1 and P63 expressing cells. P63+ cells appeared to line the basal surface of the tube, while FOXJ1 appeared to be located closer to the lumen. A cross section through a three-dimensional rendered airway revealed that the structure had a lumen lined by FOXJ1+ cells, while P63+

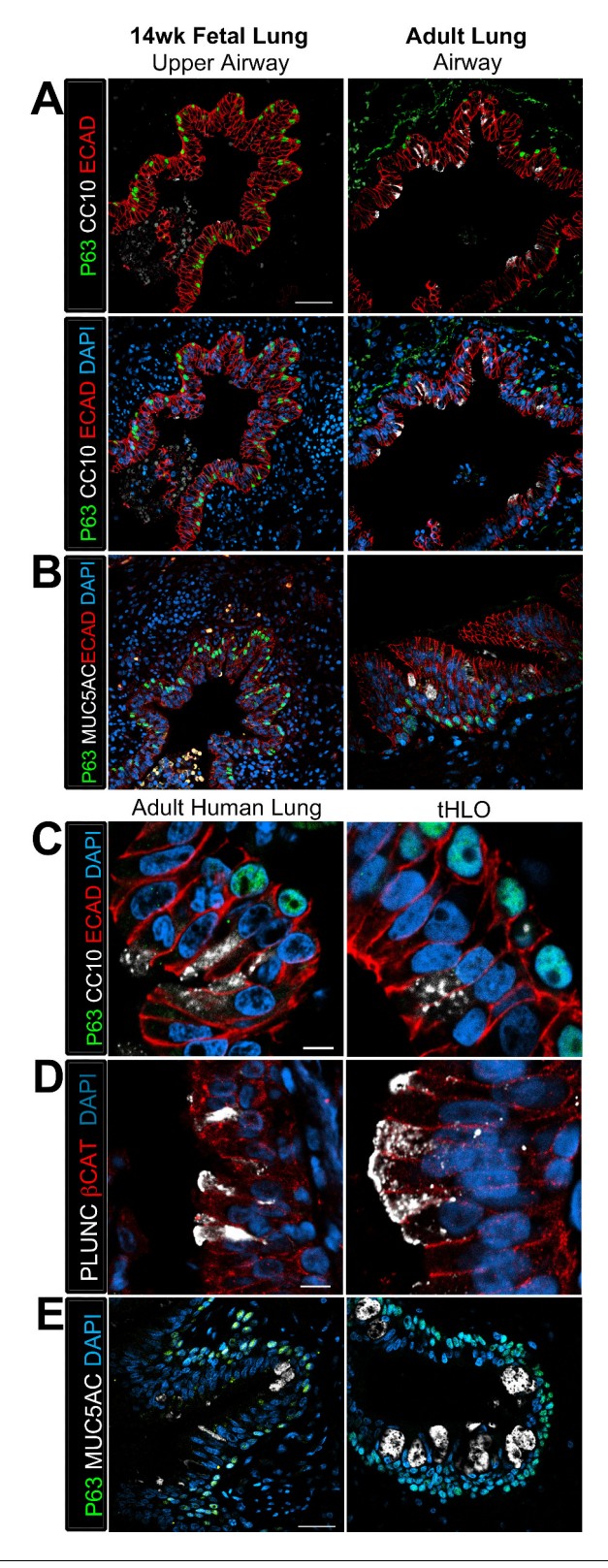

**Figure 3.** Transplanted HLO-scaffolds possessed lung secretory cells. (A–B) 14 week fetal upper airway possessed P63+ basal cells (green), but did not possess CC10+ club cells (**A**, white) or MUC5AC+ goblet cells (**B**, white) while the adult airway expressed both, CC10 (**A**) and MUC5AC (**B**). Scale bars in **A–B** represent 50 µm. (C–D) Airway-like structures in tHLOs were lined with the basal cell marker P63 (green). Some luminal cells expressed the club cell

*Figure 3 continued on next page*

*Figure 3 continued*

markers CC10 (**C**, white), or PLUNC (**D**, white), and the goblet cell marker MUC5AC (**E**, white) . Scale bars in **C–E** represent 10 µm.

The following figure supplements are available for figure 3:

**Figure supplement 1.** The club cell marker PLUNC is not detected in 14 wk fetal lungs, but is expressed in the adult lung airway epithelium.

**Figure supplement 2.** tHLOs derived from H1 and H9 hESC lines contain airway-like structures that express mature markers.

---

cells lined the outside of the tube along the basal side of the airway-like tissue. This organization was similar to the upper airways in the adult and fetal human lung (*Figure 2F*, *Figure 2—figure supplement 1A–C*). Although several tube-like structures were observed, micro-dissection was very disruptive, and did not allow accurate quantitation of the number of these structures. Thus, it is likely that not all airway-like structures were shaped as a tube, and cyst-shaped airway structures are also present within the transplants. In the adult airway and fetal upper airway sections, the majority of the cells facing toward the lumen express the ciliated cell marker FOXJ1 and are multiciliated, demonstrated by ACTTUB staining (*Figure 2G*, *Figure 2—figure supplement 1B–D*. In vitro grown HLOs have scattered cells that express FOXJ1 and few disorganized cells that are multiciliated as shown by ACTTUB immunofluorescence (*Figure 2G*, *Figure 2—figure supplement 1B,D*). In contrast, most of the cells facing toward the lumen in tHLOs were FOXJ1+, multiciliated as shown by ACTTUB immunofluorescence, and highly organized in a manner similar to the adult airway (*Figure 2G*, *Figure 2—figure supplement 1B–D*). In addition, microdissection of tHLO airway-like structures revealed abundant beating multiciliated cells suggesting that these cells are functional (*Video 1*).

In order to demonstrate variability across multiple tHLOs (n = 9 independent transplants), we quantitated the percent of all epithelial structures within tHLOs that are NKX2.1+ (*Figure 1—source data 1*), along with the percent of NKX2.1+ epithelial structures that also displayed improved organization and possessed differentiated cell types (*Figure 2H*, *Figure 2—source data 1*). We observed that 97.53% of all NKX2.1+ airway-like structures in the tHLOs possessed abundant multiciliated cells, as determined by ACTTUB immunofluorescence (*Figure 2H*, *Figure 2—source data 1*). This finding is consistent with data demonstrating that multiciliated cells are the most abundant cell type within human adult airway epithelium (*Mercer et al., 1994*).

The secretory cells of the lung, goblet and club cells, are fewer in number and are scattered throughout mature human airways (*Figure 3A–B*) (*Mercer et al., 1994*). We had previously reported that HLOs grown in vitro possessed rare cells expressing low levels of the club cell marker CC10 and we did not detect cells expressing MUC5AC protein, which is found in goblet cells (*Dye et al., 2015*). We asked if the lack of differentiated secretory cell types reflected an immature stage of development by comparing cellular differentiation in human adult and fetal airways. In adult airways, cells expressing the club cell markers CC10 and PLUNC and cells expressing the goblet cell marker MUC5AC were scattered throughout the airway (*Figure 3A–B*, *Figure 3—figure supplement 1*). In contrast, we could not find evidence for robust expression of any of these markers in the human fetal lung at 14 weeks of gestation (*Figure 3A–B*, *Figure 3—figure supplement 1*). Similar to the adult airway,

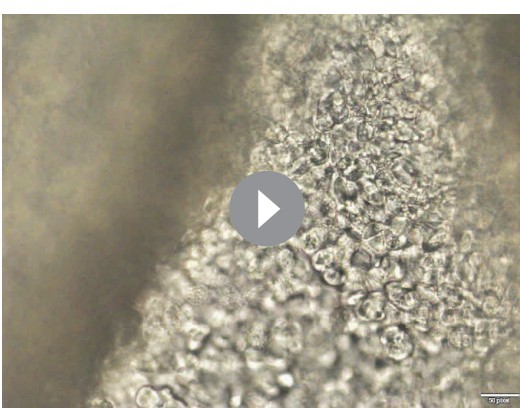

**Video 1.** Multiciliated cells had beating cilia. Videos were taken of the dissected tHLO epithelium. at various magnifications denoted on the video

we observed that the tHLO epithelium had distinct CC10+, PLUNC+, and MUC5AC+ cells (*Figure 3C–E*). Further quantification of secretory cells in multiple tHLOs (n = 9 independent transplants) revealed biological heterogeneity across samples. On average 45.55% of all NKX2.1+ airways contained CC10+ secretory cells and 70.43% contained MUC5AC+ secreting cells, however; variability ranged from some individual tHLOs possessing no secretory cells (n = 2/9 possessed no CC10+; n = 1/9 possessed no MUC5AC+) to 100% of all airway structures within an individual tHLO possessing secretory cell types (n = 1/9 CC10+; n = 3/9 MUC5AC+) (*Figure 2H*, *Figure 2—source data 1*). Highly organized airway-like structures with multiple cell types (P63+ basal, ACTTUB+ multiciliated, CC10+ club and MUC5AC+ goblet cells) were also observed in tHLOs derived from the H9 and H1 cell lines, indicating that maturation of airway-like structures upon transplantation is reproducible across multiple cell lines (*Figure 3—figure supplement 2*). Moreover, airway-like structures transplanted for different lengths of times (8 weeks, 15 weeks) appeared similar in epithelial organization and in expression of mature airway markers (*Figure 3—figure supplement 2*). Taken together, our data demonstrate that in vivo grown tHLOs possess airway-like epithelium with basal, ciliated, goblet, and club cells in a highly organized pseudostratified epithelium that resembled both the structure and cellular diversity of adult human airways.

## Transplanted HLOs possess diverse mesenchymal cell types and vasculature

Along with the epithelium, native adult airways are surrounded by smooth muscle, myofibroblasts, and cartilage. The tHLO airway structures were surrounded by cells expressing both smooth muscle actin (SMA) and PDGFRα positive (SMA+/PDGFRα+) along with SMA+PDGFRα- cells (*Figure 4A*), which are markers for myofibroblasts and smooth muscle respectively (*Boucherat et al., 2007*; *Hinz et al., 2007*; *Chen et al., 2012*). We have previously reported that HLOs grown in vitro did not possess cartilage (*Dye et al., 2015*). Consistent with these previous findings, cartilage was not present in control scaffolds seeded with HLOs and grown in vitro, as shown by the absence of SafraninO staining (*Figure 4B*). Interestingly, tHLOs possessed areas of organized cartilage indicated by cell morphology and staining for SafraninO (*Figure 1H*, *Figure 4B*). Cartilage within the tHLOs expressed the cartilage marker SOX9, along with the human mitochondrial marker, huMITO, indicating that the cartilage was derived from the transplanted tissue (*Figure 4—figure supplement 1A*). Further analysis of in vitro grown HLOs revealed that there were areas of SOX9+ mesenchyme, suggesting that a putative population of cartilage precursors is present in the HLOs grown in vitro (*Figure 4—figure supplement 1B*).

Finally, upon gross inspection of tHLOs, we observed abundant vasculature associated with the tissue, an observation that is consistent with transplanted human intestinal organoids (*Watson et al., 2014*).

## Discussion

To date, several groups have derived two and three-dimensional lung models from both primary human lung tissue and from hPSCs that consist of airway epithelial cell types (*Barkauskas et al., 2013*; *Rock et al., 2009*; *Gotoh et al., 2014*; *Konishi et al., 2016*; *Dye et al., 2015*; *Firth et al., 2014*; *Mou et al., 2016*; *Wong et al., 2012*; *Huang et al., 2013*). Tissue grown on an air-liquid interface generated an organized pseudostratified epithelium that reflects most of the cellular diversity of the human airway (*Mou et al., 2016*; *Wong et al., 2012*), while other models possess an airway epithelium with a supporting mesenchyme (*Firth et al., 2014*). Previous work has also shown that lung epithelial cells derived from hPSC monolayers transplanted underneath a mouse kidney capsule can further differentiate into cells expressing airway and alveolar markers over a period of six months (*Huang et al., 2013*). However, it was unclear how robust this approach was, and whether this yielded tissue with any degree of organization. To our knowledge, the work presented here is the first to show that hPSC-derived HLOs are able to engraft in vivo and differentiate into an organized pseudostratified airway-like epithelium with a complete repertoire of airway cell types and supporting mesenchyme, along with a vascular supply and the ability to survive for long periods of time. Interestingly, our findings demonstrated that despite the fact that in vitro grown HLOs possessed alveolar cell types, we found no evidence of alveolar cells or alveolar structure in tHLOs. Given that the alveolar cell types present in HLOs were few in number, it is possible that these cells were not

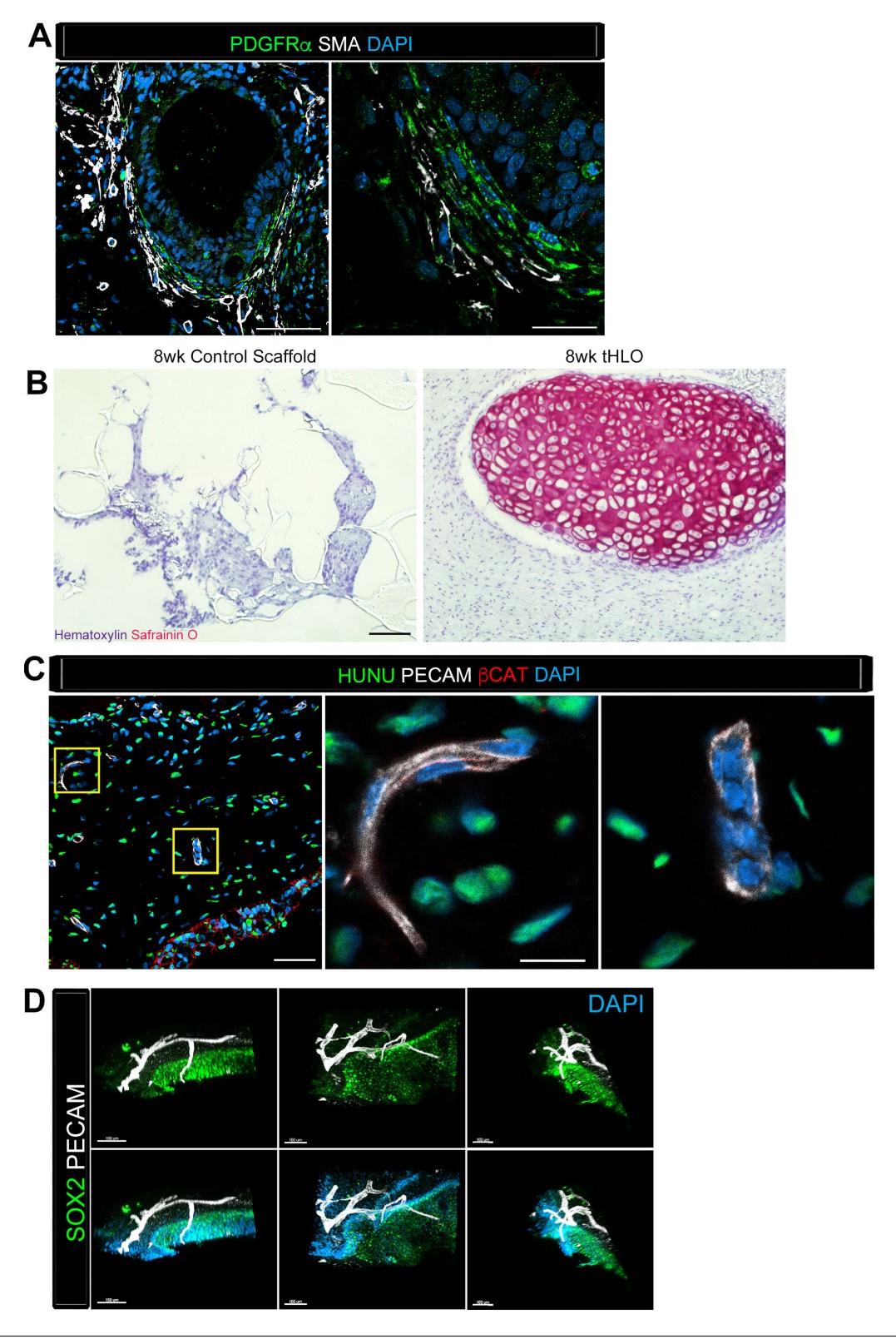

**Figure 4.** Transplanted HLO-scaffolds consisted of mesenchymal cells and vasculature. (**A**) Airway-like structures were surrounded by myofibroblasts, PDGFRα+ (green) and SMA+ (white) as well as smooth muscle, PDGFRα-/SMA+ (white only). Scale bars represent 50 µm in the lower mag image (left panel) and 25 µm in the lower mag image (right panel). (**B**) SafraninO staining showed clusters of cartilage in the 8 wk tHLO (right panel) but not in the scaffold grown in vitro for 8 weeks (left panel) Scale bar represents 100 µm. (**C**) Some βCAT+ (red) cells surrounding the airway-like structures expressed

*Figure 4 continued on next page*

*Figure 4 continued*

vasculature marker PECAM (white), but did not express human nuclear marker (HUNU, green) indicating that the vasculature is of host origin. Scale bars in A represent 50 μm lower mag image (left panel) 25 μm in bottom panel in the lower mag image (right panel). The low mag image scale bar represents 50 μm and high mag represents 10 μm. (D) 3D rendering of z-stack images on thick 12 wk tHLO sections (derived from H9 hESC) revealed the PECAM+ vascular network (white) around the SOX2+ airway epithelium (green). Scale bar represents 100 μm.

The following figure supplement is available for figure 4:

**Figure supplement 1.** Cartilage observed in transplanted HLOs is of human origin.

abundant enough to persist in vitro, or alternatively, that PLG scaffolds did not promote their survival. Ongoing studies are aimed at better understanding how to generate these distal airway structures.

Here, we used microporous PLG scaffolds to provide a physical environment for HLO engraftment in vivo however; alternative engineering approaches have also been developed as a platform to grow organized lung-like structures. For example, decelluarized lung scaffolds have been employed to seed primary and derived lung tissue in order to provide an appropriate physical and structural environment and in attempts to generate lung-like tissue (*Dye et al., 2015*; *Gilpin et al., 2014*, *2016*; *Booth et al., 2012*; *Ghaedi et al., 2014*). Many efforts have focused on seeding alveolar tissue including alveolar type I and II cells onto acellular matrices (*Gilpin et al., 2014*; *Booth et al., 2012*; *Ghaedi et al., 2014*), and lung tissue derived from mouse embryonic stem cells seeded on decellularized rat lung matrix enhanced maturation of airway cell types and epithelial organization (*Shojaie et al., 2015*). Our data previously demonstrated that in vitro grown HLOs seeded onto an adult human decelluarized lung matrix allowed adherence of the HLO epithelium to the airway matrix and resulted in the differentiation of a small number of multiciliated cells, but did not enhance club or goblet cell differentiation (*Dye et al., 2015*). Collectively, our data along with the data from decelluarized lung scaffolds suggest that providing an ideal physical environment may be an important factor for promoting tissue organization and differentiation, but that additional cues will also be required for full tissue maturation.

Recent breakthroughs in biomimetic microfluidic cell-culture devices called 'organs on a chip' have also generated airway epithelium from human adult airway cells grown on an air-liquid interface platform. The microfluidic devices allow are designed to mimic the microenvironment of the airway, and can include controlled media flow and multiple cell layers including human primary epithelium, fibroblasts, and endothelial cells (*Sellgren et al., 2014*; *Benam et al., 2016*). These in vitro systems successfully mimic the microenvironment of an airway, but are designed for small scale and high throughput applications. In contrast tHLOs are an in vivo system that may be more appropriate for low-throughput disease modeling or for pre-clinical drug testing, which has a great unmet need for

better models that more accurately mimic human tissue in both healthy and diseased states in order to improve prediction of drug efficacy. This is particularly true for diseases that affect the lung, in part because commonly used animal models do not faithfully recapitulate several key aspects of the human disease (*Moore and Hogaboam, 2008*). Moreover, there is generally a poor track record of accurately predicting which drugs will work in humans based on pre-clinical animal models (*Ruggeri et al., 2014*; *Mak et al., 2014*), as up to 80% of drugs that pass pre-clinical tests fail in humans (*Perrin, 2014*). A prime example of this is in lung disease where many therapies have shown benefit in mouse models, but only 2 of these have shown benefit in humans (*Myllärniemi and Kaarteenaho, 2015*).

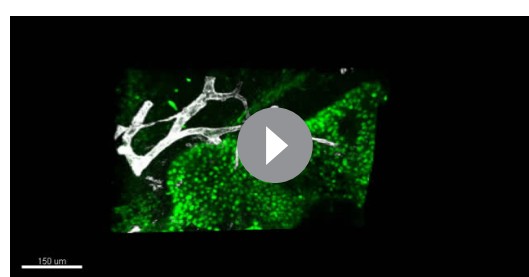

**Video 2.** SOX2+ tHLO airway is surrounded by PECAM+ vasculature. 3D rendering of z-stack images on thick 12 wk tHLO sections with a 360° rotation revealed the PECAM+ vascular network (white) around the SOX2+ airway epithelium (green). Still images of this video are shown in *Figure 4D*. Scale bar represents 200 μm.

**Table 2.** Primary and secondary antibody information.

| Primary antibody | Source | Catalog # | Dilution | Clone |
|---|---|---|---|---|
| Goat anti-β-Catenin (βCAT) | Santa Cruz Biotechnology | sc-1496 | 1:200 | C-18 |
| Goat anti-CC10 | Santa Cruz Biotechnology | sc-9770 | 1:200 | C-20 |
| Goat anti-SOX9 | R&D Systems | AF3075 | 1:500 | polyclonal |
| Goat anti-VIMENTIN (VIM) | Santa Cruz Biotechnology | sc-7558 | 1:100 | S-20 |
| Mouse anti-Acetylated Tubulin (ACTTUB) | Sigma-Aldrich | T7451 | 1:1000 | 6-11B-1 |
| Mouse anti-E-Cadherin (ECAD) | BD Transduction Laboratories | 610181 | 1:500 | 36/E-Cadherin |
| Mouse anti-FOXJ1 | eBioscience | 14-9965-82 | 1:500 | 2A5 |
| Moues anti- Human Nuclear Antigen** | Millipore | MAB1281 | 1:200 | monoclonal |
| Mouse anti- Human Mitochondria (huMITO) | Millipore | MAB1273 | 1:500 | 113-1 |
| Mouse anti-PLUNC | R&D Systems | MAP1897 | 1:200 | monoclonal |
| Rabbit anti-Cytokeratin5 (CK5) | Abcam | ab53121 | 1:500 | polyclonal |
| Rabbit anti-NKX2.1 | Abcam | ab76013 | 1:200 | EP1584Y |
| Rabbit anti-P63 | Santa Cruz Biotechnology | sc-8344 | 1:200 | H-129 |
| Rabbit anti-PDGFRα | Santa Cruz Biotechnology | sc-338 | 1:100 | C-20 |
| Rabbit anti-SOX2 | Seven Hills Bioreagents | WRAB-SOX2 | 1:500 | polyclonal |
| Rat anti-PECAM-1 | BD Biosciences | 557355 | 1:200 | monoclonal |
| Biotin-Mouse anti MUC5AC* | Abcam | ab79082 | 1:100 | monoclonal |
| Cy3- Mouse anti Actin-alpha smooth muscle (SMA)* | Sigma | C6198 | 1:400 | monoclonal |
| **Secondary antibody** | **Source** | **Catalog #** | **Dilution** | |
| Donkey anti-goat 488 | Jackson Immuno | 705-545-147 | 1:500 | |
| Donkey anti-goat 647 | Jackson Immuno | 705-605-147 | 1:500 | |
| Donkey anti-goat Cy3 | Jackson Immuno | 705-165-147 | 1:500 | |
| Donkey anti-mouse 488 | Jackson Immuno | 715-545-150 | 1:500 | |
| Donkey anti-mouse 647 | Jackson Immuno | 415-605-350 | 1:500 | |
| Donkey anti-mouse Cy3 | Jackson Immuno | 715-165-150 | 1:500 | |
| Donkey anti-rabbit 488 | Jackson Immuno | 711-545-152 | 1:500 | |
| Donkey anti-rabbit 647 | Jackson Immuno | 711-605-152 | 1:500 | |
| Donkey anti-rabbit Cy3 | Jackson Immuno | 711-165-102 | 1:500 | |
| Donkey anti-goat 488 | Jackson Immuno | 705-545-147 | 1:500 | |
| Donkey anti-goat 647 | Jackson Immuno | 705-605-147 | 1:500 | |
| Donkey anti-goat Cy3 | Jackson Immuno | 705-165-147 | 1:500 | |
| Donkey anti-mouse 488 | Jackson Immuno | 715-545-150 | 1:500 | |
| Donkey anti-mouse 647 | Jackson Immuno | 415-605-350 | 1:500 | |
| Donkey anti-mouse Cy3 | Jackson Immuno | 715-165-150 | 1:500 | |
| Donkey anti-rabbit 488 | Jackson Immuno | 711-545-152 | 1:500 | |
| Donkey anti-rabbit 647 | Jackson Immuno | 711-605-152 | 1:500 | |
| Donkey anti-rabbit Cy3 | Jackson Immuno | 711-165-102 | 1:500 | |

It is interesting to speculate that tHLOs will have utility as a novel model to study complex tissue-tissue interactions in airway homeostasis or disease. For example, since tHLOs have the potential to interact with the host immune system, and since the epithelium possesses diverse cell types including club and goblet cells, it may be possible to model inflammatory insults resulting in goblet cell hyperplasia. Future directions will be aimed at exploiting this new in vivo model of the human airway to better understand disease and to improve human health.

## Materials and methods

### Cell lines, human tissue and animals

Human ES line UM63-1 (NIH registry #0277) was obtained from the University of Michigan and human ES line H9 (NIH registry #0062) and H1 (NIH registry #0043) was obtained from the WiCell Research Institute. All experiments using human ES cells were approved by the University of Michigan Human Pluripotent Stem Cell Research Oversight Committee. ES cell lines are routinely karyotyped to ensure normal karyotype, at which time the sex chromosomes of each line are confirmed (H9 - XX; H1 - XY; UM63-1 - XX). Monthly mycoplasma monitoring is conducted on all cell lines using the MycoAlert Mycoplasma Detection Kit (Lonza). *Human tissue:* Normal, de-identified human fetal lung tissue was obtained from the University of Washington Laboratory of Developmental Biology. Normal, de-identified human adult lung tissue was obtained from deceased organ donors through the Gift of Life, Michigan. All research with human tissue was approved by the University of Michigan institutional review board. Animal use: All mouse work was reviewed and approved by the University of Michigan Committee on Use and Care of Animals.

### Maintenance of hESCs and generation of foregut spheroids and HLOs

Stem cells were maintained on hESC-qualified Matrigel (Corning, Cat#: 354277) in mTesR1 medium (STEM CELL Technologies). HESCs were passaged as previously described (*Spence et al., 2011*). HLOs were generated as previously described (*Dye et al., 2015*). All HLOs imaged and quantifications of tHLOs were derived from UM63-1 unless otherwise stated.

### Kidney capsule and omentum transplants

Mice were anesthetized using 2% isofluorane. The left flank was sterilized using Chlorhexidine and isopropyl alcohol. A left flank incision was used to expose the kidney. 35d HLOs were manually placed in a subcapsular pocket of the kidney of male 7–10 week old NOD-scid IL2Rgnull (NSG) mice using forceps. An intraperitoneal flush of Zosyn (100 mg/kg; Pfizer Inc.) was administered prior to closure in two layers. The mice were sacrificed and transplant retrieved after 4 weeks. Foregut spheroids mixed with 100% Matrigel were injected underneath the mouse kidney capsule using a flexible catheter and transplants were retrieved after 6 weeks. For omental transplants, the abdomen was prepped and a midline incision was used to expose the greater omentum. 65d HLOs were sutured into the greater omentum using non-absorbable suture. Transplants were retrieved after 12 weeks.

### Scaffolds transplants

PLG scaffolds were generated as previously described (*Blomeier et al., 2006*). Briefly, 2.5 mg of 6% PLG and 75 mg of NaCl (250–425 μm diameter) were mixed and incubated at 37°C for 7 min and then set at room temperature for 6 min. The mixture was pressed into 2 mm thick, 5 mm wide cylinders at 1500 psi for 30 s. Pressed scaffolds were foamed at 800 psi $CO_2$ for 16 hr. Foamed scaffolds can be stored in a dry environment for weeks. PLG scaffolds were leeched in $ddH_2O$ for two 1-hr washes to remove NaCl. The scaffolds were then washed in 70% EtOH for 2 min for sterilization prior to use in experiments and dried for 5 min. PLG scaffolds were flushed with 15 μL of cold hESC qualified Matrigel diluted in DMEM/F12 (dilution factor is lot dependent, Corning Cat#: 354277) and incubated at 37°C for 15 min. This step was repeated twice. Lung spheroids were mixed with 100% Matrigel (Corning, Cat#: 354234) and pipetted into the scaffold. The seeded scaffolds were incubated for 15 min at 37°C and then overlaid with HLO media (500 ng/mL FGF10 and 1% FBS in basal foregut media described in [*Dye et al., 2015*]). Scaffolds were cultured for 5 to 7 days in vitro with media being changed every other day. Scaffolds cultured with spheroids were removed from culture and were then immersed in Matrigel supplemented with 500 ng/mL FGF10, allowed to solidify for 5 min or were directly used for transplants. Mice were anesthetized and prepped as for omental transplants. The epididymal fat pads of male 7–10 week old NOD-scid IL2Rgnull (NSG) were exposed using a lower midline incision. Matrigel coated scaffolds were then placed along the epididymal blood vessels and covered with epididymal fat. An intraperitoneal flush of Zosyn (100 mg/kg; Pfizer Inc.) was administered after which the incision was closed in 2-layers using absorbable suture. Mice were sacrificed between 4 and 15 weeks post-transplant.

## Spheroid dye

Vybrant DiO Cell-Labeling Solution (ThermoFisher Scientific) was used based on manufacture's protocol. Briefly, spheroids were incubated with the dye (5 µL dye per 1 mL media) for 20 min at 37°C. Spheroids were washed in media two times and were ready to use and image.

## Tissue sectioning, immunohistochemistry and imaging

For human tissue sections: Anatomically, we considered the upper airway to be anything above the terminal bronchioles in the lung. Analyses conducted on human adult airways were carried out on sections through the lower bronchi of the upper airway, defined by the lack of adjacent cartilage and pseudostratified, undulated epithelial architecture. Analysis of human fetal airways was carried out on sections through bronchi and/or bronchioles of the upper airway at a level where the adjacent cartilage was present. Immunostaining was carried out as previously described (*Rockich et al., 2013*). Antibody information and dilutions can be found in *Table 2*. All images and videos were taken on a Nikon A1 confocal microscope or an Olympus IX71 epifluorescent microscope. Imaris software was used to render Z-stack three-dimensional images.

## Quantification

ECAD+ tissue was counted as airways with or without NKX2.1. In *Figure 2H*, NKX2.1+ airway-like tissue within individual tHLOs were quantitated to determine if they possessed ciliated cells, club cells or goblet cells by co-immunostaining with NKX2.1, ACTTUB, CC10 and MUC5AC antibodies, respectively. Cell height was measured of the cells facing in toward a lumen of an airway stained with ECAD by using ImageJ software.

## Experimental replicates and statistics

All immunofluorescence (IF) were done on at least three (n = 3) independent biological samples per experiment except for foregut spheroids injected into the kidney capsule. Only 1 out 3 transplanted foregut spheroids under the kidney capsule yielded retrievable tissue; therefore only 1 sample was used for the IF analysis. For the airway quantification and cell height quantification the error bars represented SEM while the long bar represented the average. Statistical differences were assessed with Prism software using multiple t tests.

## Acknowledgements

JRS is supported by the NIH-NHLBI (R01 HL119215). AJM is supported by the NIH Cellular and Molecular Biology training grant at Michigan (T32 GM007315), and by the Tissue Engineering and Regeneration Training Grant (DE00007057-40). BRD is supported by a Graduate Fellowship from the University of Michigan Rackham graduate school. The University of Washington Laboratory of Developmental Biology was supported by NIH Award Number 5R24HD000836 from the Eunice Kennedy Shriver National Institute of Child Health & Human Development.

## Additional information

### Funding

| Funder | Grant reference number | Author |
|---|---|---|
| University of Michigan | Rackham Graduate Fellowship | Briana R Dye |
| University of Michigan | Cellular and Molecular Biology Training Grant - T32 GM007315 | Alyssa J Miller |
| University of Michigan | Tissue Engineering and Regeneration Training Grant - DE00007057-40 | Alyssa J Miller |
| National Heart, Lung, and Blood Institute | RO1 HL119215 | Jason R Spence |

The funders had no role in study design, data collection and interpretation, or the decision to submit the work for publication.

## Author contributions

BRD, PHD, Conception and design, Acquisition of data, Analysis and interpretation of data, Drafting or revising the article; AJM, MSN, Acquisition of data, Analysis and interpretation of data, Drafting or revising the article; ESW, LDS, Drafting or revising the article, Contributed unpublished essential data or reagents; JRS, Conception and design, Analysis and interpretation of data, Drafting or revising the article

## Author ORCIDs

Jason R Spence, http://orcid.org/0000-0001-7869-3992

## Ethics

Animal experimentation: All work using human pluripotent stem cells was approved by the University of Michigan Human Pluiripotent Stem Cell Research Oversight Committee (HPSCRO, application #1054). All human tissue used in this work was falls under NIH Exemption 4. The tissue was not obtained from living individuals, and was de-identified. Since this work falls under NIH Exemption 4, it was given a "not regulated" status by the University of Michigan IRB (protocol # HUM00093465 and HUM00105750). All animal experiments were approved by the University of Michigan Institutional Animal Care and Use Committee (IACUC; protocol # PRO00006609).

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
