## [Decision Letter]

Thank you for submitting your article "A bioengineered niche promotes in vivo engraftment and maturation of pluripotent stem cell derived human lung organoids" for consideration by *eLife*. Your article has been reviewed by two peer reviewers, and the evaluation has been overseen by a Reviewing Editor and Janet Rossant as the Senior Editor. The reviewers have opted to remain anonymous.

The reviewers have discussed the reviews with one another and the Reviewing Editor has drafted this decision to help you prepare a revised submission.

Summary:

In this research advance, building on their previous generation of lung organoids from human pluripotent cells, the authors have described a bioengineering approach to derive airway epithelial cell structures in vivo using scaffolds and iPS derived cells. Using a combination of organoid culture with matrigel and scaffold, the layered epithelium they achieve in vivo closely parallels human airway. The imaging and especially the polarity and organization of the cell types they have achieved is quite remarkable. This technique, if as reproducible and efficient as reported, could advance the field for the study of lung disease.

Essential revisions:

The main concern raised by the reviewers was around the reproducibility of the process and the quantitation of the extent of the differentiated structures produced. There is still wide variability in success for each transplant. What parameters are critical in determining efficiency? If this approach is to become a major tool for studying lung function and lung disease, the robustness with which the complex structures can be produced in different assays and using different starting cell lines is key.

The authors are requested to consider these issues and address them as outlined in the reviewers' comments below:

A major concern is that the overall reproducibility of the PLG technique is not clear. For example, it is not clear if every region of NKX2.1+,ECAD+ epithelium stained at 8 weeks post-transplant was differentiated as airway, or if differentiation was more patchy. Please quantify appropriately to clarify this point. In addition, 3x 15 week transplants are listed in Table 1 as showing NKX2.1+ regions with airway-like differentiation. Were these identical to the 8 week transplants? Two different cell lines are listed as being used in Table 1, were the results identical between them?

Similarly, was the airway epithelium obtained at 8-15 weeks post-transplant always arranged in tubes as shown in Figure 2?

In general quantitation is appropriate and well described (although as noted above the origin of the samples scored is not shown e.g. all H9, or all UM63-1, or a mixture? 8 weeks or 15 weeks?) e.g. cell height of airway epithelium Figure 2.

However, the quantitation of the secretory cell phenotype is missing (Figure 3). It may be difficult to quantitate accurately due to the low numbers of these cell types in the human airways. At the very least the numbers of samples in which CC10+, MUC5AC+ and PLUNC+ cells can be identified should be shown.

Related to this point, how are the fetal airways classified as upper or lower? And in which region of the adult airways are the images of secretory cells shown? e.g. mid-size bro.

---

## [Author Response]

*Essential revisions:*

*The main concern raised by the reviewers was around the reproducibility of the process and the quantitation of the extent of the differentiated structures produced. There is still wide variability in success for each transplant. What parameters are critical in determining efficiency? If this approach is to become a major tool for studying lung function and lung disease, the robustness with which the complex structures can be produced in different assays and using different starting cell lines is key.*

We wholeheartedly agree with the overarching concern raised by the reviewers that reproducibility, and variability across transplants, are important considerations. Such a concern is very valid in a field where variability is often not addressed adequately (or at all in some cases), and we appreciate the opportunity to improve our manuscript with additional data to help paint a transparent picture of the transplanted lung organoid system that we are describing. We will note that for the revision, we have added transplant from another cell, bringing the number of different cell lines used to three (H1, H9, UM63-1), (Table 1). While the absolute number of airway-like epithelium structures varied widely amongst individual transplants ([Supplementary-material SD1-data]), we will note that 100% of transplants examined from all three cell lines possessed at least some airway-like epithelium. Thus, we feel that using three different cell lines, and a total of n=19 individual constructs transplanted into different mice for at least 8 weeks, with 100% of these possessing some airway-like epithelium, that we have rigorously tested the robustness of this system.

*The authors are requested to consider these issues and address them as outlined in the reviewers' comments below:*

*A major concern is that the overall reproducibility of the PLG technique is not clear. For example, it is not clear if every region of NKX2.1+,ECAD+ epithelium stained at 8 weeks post-transplant was differentiated as airway, or if differentiation was more patchy. Please quantify appropriately to clarify this point.*

We have now quantified n=9 individual transplants derived from the UM63-1 cell line (Figure 2, [Supplementary-material SD2-data]). Note that at the time of generating data for revisions, we had tissue sections left from only 9 transplanted HIOs (out of 12 total). To conduct the quantitation, we did co-staining for NKX2.1/ACTTUB, NKX2.1/CC10 and NKX2.1/MUC5AC. We then calculated percent of NKX2.1+ epithelial structures that were also positive for the co- stained markers. In the plotted data (Figure 2), each individual transplanted construct is presented as a different color, so that the reader can get an idea of the percent of co-staining that was present in each transplant. Since multiciliated cells are abundant in human airways, we only considered an NKX2.1+ structure to be an airway-like structure when most/all of NKX2.1+ region contained multiciliated cells indicated by ACTTUB staining (appearing similar to staining in Figure 2—figure supplement 1). Co-staining with ACTTUB and NKX2.1 revealed that 61 of 63 airways counted in 9 constructs had 100% of airway-like structures with co-staining of NKX2.1/ACTTUB similar to that shown in confocal images throughout the manuscript. As the reviewers suggested (see below), secretory cells were more rare, and were observed in a more variable fashion (Figure 2, [Supplementary-material SD2-data]). Nonetheless, this analysis supports that individual NKX2.1+ epithelial structures within tissue sections are not “patchy” – that they largely resemble mature airways and are populated with abundant multiciliated cells, and to a lesser extent, also possess secretory cells.

*In addition, 3x 15 week transplants are listed in Table 1 as showing NKX2.1+ regions with airway-like differentiation. Were these identical to the 8 week transplants? Two different cell lines are listed as being used in Table 1, were the results identical between them?*

We have also now added a cohort of transplanted HLOs derived from the H1 cell line, which were harvested after 8 weeks. Examining the airway-like structures from three different lines, and from different times transplanted in vivo(summarized in Table 1) we find that airway-like structures are similar, regardless of transplant time or cell line. To support this claim, we now include representative images from H9-derived tHLOs transplanted for 15 weeks, and from H1-derived tHLOs transplanted for 8 weeks (Figure 3—figure supplement 2).

*Similarly, was the airway epithelium obtained at 8-15 weeks post-transplant always arranged in tubes as shown in Figure 2?*

When micro-dissecting tHLOs, we routinely find tube-like structures present, however; we have not rigorously evaluated the extent that ‘tube-like’ structures are present across different transplant times. Moreover, we have not quantified ‘tube-like’ structures relative to ‘cyst-like’ structures. Thus, while we have chosen to keep the ‘tube-like’ data in the manuscript, we have now softened our claims, and have added the above caveats to this observation in the text.

*In general quantitation is appropriate and well described (although as noted above the origin of the samples scored is not shown e.g. all H9, or all UM63-1, or a mixture? 8 weeks or 15 weeks?) e.g. cell height of airway epithelium Figure 2.*

Thank you for pointing out this detail. We have now explicitly clarified which transplants were used for all analyses. In general, all analyses are carried out with UM63-1 at 8 weeks, unless otherwise noted. We chose to use this data because we had the largest transplant cohort from this cell line and felt that the increased sample size would yield more powerful quantitative data.

*However, the quantitation of the secretory cell phenotype is missing (Figure 3). It may be difficult to quantitate accurately due to the low numbers of these cell types in the human airways. At the very least the numbers of samples in which CC10+, MUC5AC+ and PLUNC+ cells can be identified should be shown.*

This quantitation (number of NKX2.1+ epithelial structures in which CC10+ and MUC5AC+ cells are present) is now included in Figure 2 and [Supplementary-material SD2-data].

Related to this point, how are the fetal airways classified as upper or lower? And in which region of the adult airways are the images of secretory cells shown? e.g. mid-size bro.

We have now added a detailed description of the areas in the lungs from which we took images for both fetal and adult tissues, which can be found in the methods section. Following discussions with Dr. White, our clinical collaborator, for purposes of this manuscript, we are defining upper airways as anything above the terminal bronchioles in the lungs. Given that smaller bronchioles have fewer secretory cells, and given that we did not see evidence for secretory cells in the fetal lung tissue we examined, we confirmed that the fetal lung images we used were from upper airways in which cartilage condensations are present. We feel that this landmark (cartilage) ensures that we are in an area within the upper airway that will eventually give rise to secretory cells.